# A Review of the Extraction and Closed-Loop Spray Drying-Assisted Micro-Encapsulation of Algal Lutein for Functional Food Delivery

Zexin Lei *[ID] and Timothy Langrish [ID]

Drying and Process Technology Group, School of Chemical and Biomolecular Engineering Building J01, The University of Sydney, Darlington, NSW 2006, Australia; timothy.langrish@sydney.edu.au
* Correspondence: zlei7127@uni.sydney.edu.au; Tel.: +61-450588867

**Abstract:** In this study, the physical and chemical properties and bioavailability of lutein have been summarized, with the novelty of this work being the review of lutein from production to extraction, through to preservation and drying, in order to deliver a functional food ingredient. The potential health functions of lutein have been introduced in detail. By comparing algae and marigold flowers, the advantages of algae extraction technology have been discussed. In this article, we have introduced the use of closed-loop spray drying technology to microencapsulate lutein to improve its stability and solubility. Microencapsulation of unstable substances by spray drying is a potentially useful direction that is worth exploring further.

**Keywords:** lutein; algae; extraction; closed loop spray drying; microencapsulation; glass transition temperature

## 1. Lutein Introduction

Lutein is mainly found in fruits and vegetables in the form of pigments. It is an antioxidant that has a potential role in preventing or reducing age-related macular degeneration, so lutein supplementation might be considered as a functional food. The structure of lutein ((3R,3R,6R)β,ε-carotene-3-3-diol) consists of a long carbon chain structure with alternating single and double bonds. There is a cyclic vinyl structure at both ends of the carbon skeleton, which is a characteristic structure of carotenoids. The nine double bonds it contains specifically absorb blue light, which makes lutein yellow [1].

From a functional food perspective, lutein cannot be synthesized by the human body, so bioavailability is important after ingesting lutein from external sources [2]. Kurilich et al. reported that immunolabeling was used to find that lutein existed in the blood for several hours after eating lutein-containing foods [3]. Tyssandier et al. did the same experiment and found that low levels ($4.7 \pm 0.1$ nmol/L) of lutein were metabolized from the diet [4]. The different results from the two experiments may be related to whether the diet eaten in the experiment contains more lipids. Zaripheh and Erdman Jr. reported that carotenoids are consistent with lipid metabolism channels, so solubility may be an important factor affecting bioavailability [5]. It is known that carotenoids have antioxidant properties, which are involved in the elimination of free radicals and peroxides in the body. Marchetti et al. have added dried nettle leaves to egg pasta to make it rich in lutein and β-carotene, creating a functional food [6].

However, the technology for providing the lutein as a supplement is not trivial. Currently, lutein is extracted from marigold flowers in industry. As the market demand for lutein continues to increase, the existing technology can no longer meet production needs, so *Chlorella* as a possible source of lutein has been shown to contain more lutein than marigold [7]. Therefore, reviewing the technology used for the extraction and processing of lutein from algae is relevant to the functional food industry.

Recently, Becerra et al. have reviewed the use of lutein as a functional food ingredient focusing on its stability and bioavailability [8]. This paper discussed the chemistry, the healthy benefits, and extraction of lutein in considerable detail. The drying and encapsulation steps of the lutein and emulsion-based delivery systems have been given less attention in this previously published work. This current paper discusses closed-loop spray drying, microencapsulation, and emulsion-based delivery systems, and we discuss the essential nature of using this system for the solvents that are commonly required for lutein extraction. The microencapsulation section summarizes the relevant technology for lutein and analyzes mass balances in the spray drying process. The emulsion section introduces several common lipophilic emulsion matrix models and their role in improving bioavailability and inhibiting chemical degradation. A review of lutein health benefits, together with an integrated view of the extraction of lutein, its drying, and microencapsulation, has not been presented in the previous literature, so this review and gap analysis are novel contributions of this work.

## 2. Benefits of Lutein

Carotenoids, especially lutein, play a key role in human health, especially in the eyes through their antioxidant properties. The following sections summarizes the benefits of lutein.

### 2.1. Antioxidant

Lycopene and lutein are highly effective lipid peroxide scavengers. Broniowska et al. have done an in-depth study of their antioxidant efficiency. Lutein can significantly reduce the rate of formation of MDA in liposomes, and the content of MDA reflects the degree of peroxidation of the cell membrane [9].

### 2.2. Anti-Cancer

Reduced DNA activity and oxidative damage are some direct causes of cancer, so substances that contribute to antioxidants have the potential to contribute to cancer reduction. Toniolo et al. reported that insufficient vitamin supplementation may increase the risk of breast cancer [10]. Chew, Brown, Park, and Mixter [11]. Chew, Brown, Park, and Mixter reported that lutein with an edible content of 0.002% in the blood can inhibit the growth of cancer cells by selectively regulating apoptosis.

### 2.3. Eye Disease Prevention

Although the importance of vitamin A for vision has been recognized for many years, it has been found that vitamins, carotenoids, and trace elements, especially lutein in foods such as fruits and vegetables, are important for the eye. The yellow color of the human retina macular is due to the presence of macular pigments. The degradation products of lutein and the geometric isomers of lutein are found in the retina. It has been suggested that these macular carotenoids play a role in preventing macular degeneration. Age-related macular degeneration (AMD) is an irreversible process, which is the main cause of blindness and occurs as the incidence of damage increases. By absorbing blue light, the macular pigment protects the underlying photoreceptor layer from photodamage [12]. The risk of developing AMD may be affected by diet, low levels of lutein in serum or retina, and excessive exposure to blue light.

### 2.4. Application to Cardiovascular Diseases

The protective effect of antioxidants is well known, which may help to reduce certain cardiovascular diseases [13]. The effects of lutein on the oxidation of low-density lipoprotein (LDL) and atherosclerosis have been studied. Nicolle et al. reported the antioxidant effects of carotenoids in the diet, and the potential roles in preventing degenerative diseases were studied such as atherosclerosis [14]. Nicolle et al. also reported that incorporating car-

rots into the rat diet improves cholesterol absorption and bile acid excretion and increases antioxidant levels, which helps to protect the cardiovascular system [14].

## 3. Extraction of Lutein from Algae

There are several considerations in the choice of algae as a source of lutein by extraction, the content of lutein and the production rate of the algae. A high content of lutein means that a high yield gives easier industrial production. The higher the yield of algae, the lower the cost. In addition, whether the cell wall is easy to destroy or not is an important question.

### 3.1. Selection of Algae

The selection of algae is based on some further considerations, such as the growth rate, the lutein/zeaxanthin ratio, the chlorophyll A /lutein ratio, and the lutein content. A high growth rate means that green algae synthesize more lutein per unit time. Choosing a species with a high lutein content will help improve the effectiveness of lutein separation. Similarly, high cell density and high biomass per unit volume also mean high lutein production. Table 1 summarizes the lutein content in different microalgae as reported by McClure et al.

**Table 1.** Summary of published studies examining lutein production using microalgae, adapted from Ref. [15].

| Species | Maximum Specific Lutein Concentration (mg g$^{-1}$ DCW) | Biomass Concentration (g L$^{-1}$) | Lutein Productivity (mg L$^{-1}$ day$^{-1}$) |
|---|---|---|---|
| *Chlorella Minutissima* | 8.24 | 3 | 6.4 |
| *Chlorella protothecoides* | 4.58 | 19.6 | 11.3 |
| *Chlorella sorokiniana* | 5.21 | 2.5 | 5.78 |
| *Chlorella vugaris* | 3.86 | 1.28 | 0.51 |
| *Chlorella vugaris (UTEX 1803)* | 9.82 | 2.93 | 11.98 |
| *Chlorella vugaris (CS-41)* | 4.85 | 16.4 | 8.4 |

To reduce the moisture level in the algae cells from 99.5% to 75%, centrifugation, gravity sedimentation, membrane filtration, and other mechanical means may be used. In addition, separation can be enhanced by an additional coagulant, such as poly-aluminum in combination with chloride and chitosan, or by adjusting the pH. Cell walls can be broken by mechanical means, such as bead milling, ultrasonication, hydrodynamic cavitation, and homogenization. Non-mechanical methods include physical, chemical, and biological processes. The moisture content of chlorella after drying has been reported to be ~10%. Figure 1 shows the structure of chlorella cell [16].

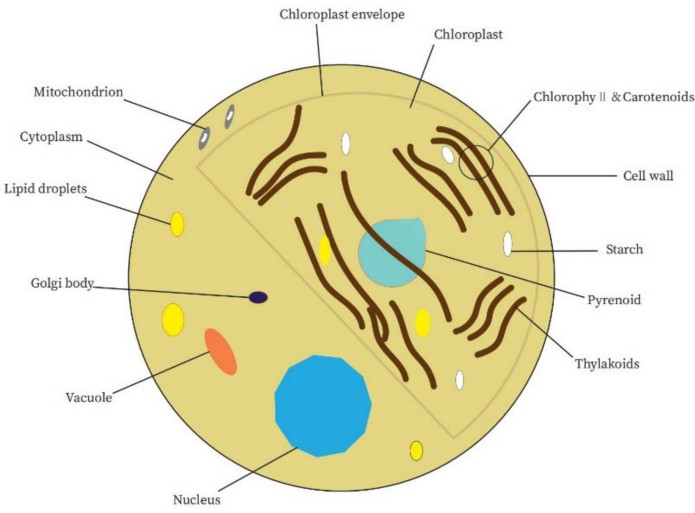

**Figure 1.** Structure of *Chlorella vulgaris*, adapted from Ref. [16].

### 3.2. Cell Disruption

Making the cell wall of chlorella more permeable is a direct factor affecting the yield of the extract. At present, the main methods for improving permeability include manual grinding, ultrasonication, bead milling, and enzymatic lysis [17]. The most convenient and cheap method at the laboratory scale is manual grinding. Liquid nitrogen can be added to the sample to increase the brittleness of cells so that the cell walls are easy to crack. Adding some quartz can increase friction, which makes cell walls break more completely [18]. Compared with manual grinding, bead milling is a more efficient way to break the cell wall. The bead milling method uses the shearing force between the solids to break the cell. It is a very effective method of physical cell disruption. The grinding chamber may be equipped with steel balls or small glass balls to improve the grinding capacity [19]. Compared with physical methods, enzymatic lysis is more thorough in breaking cell walls, but it also needs to be operated in a specific environment. The optimum temperature of cellulase is 55 °C, the pH value is 4.8, and it needs to be placed in a water bath for 10 h [17].

Prabakaran et al. tested different disruption methods for disrupting *C. vulgaris* (Figure 2) [20]. The lipid contents of cells after disruption were used to indicate the extent of disruption. The higher the degree of damage, the more lipid that the cells release. It was concluded that the maximum content of lipids may be obtained by autoclaving and microwaving. The lowest lipid concentration was obtained by autoclaving. Compared with the three varieties, *Chlorella* has the highest content of lutein. Pernet and Tremblay obtained a similar result, that different disruption methods affect the TAG levels extracted from *Chaetoceros gracilis*, and liquid nitrogen grinding is an effective method. Because frozen cells will crack with low impacts at very low temperatures (−196 °C), the liquid nitrogen will evaporate after cell disruption and will not damage the extraction of lipid in the next step. Low temperatures can also prevent lipid from being oxidized, thus improving the production of lipids. Pernet and Tremblay's study also reported that manual grinding and ultrasonication can break the cell walls, but the resulting lipid contents are low [21].

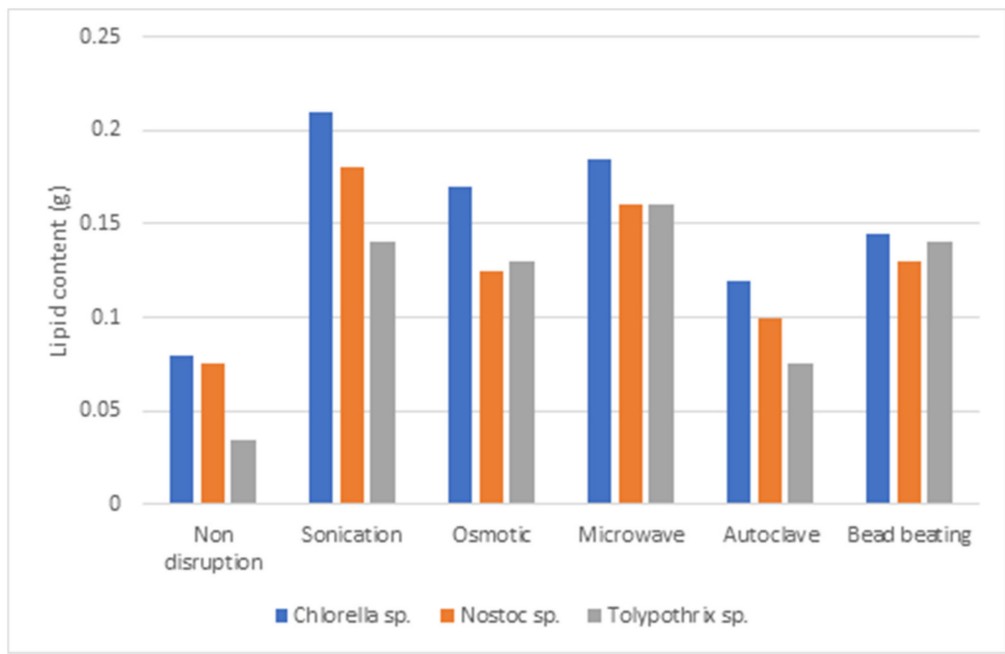

**Figure 2.** Lipid extraction efficiency according to microalgae species and method, adapted from Ref. [20].

Geciova, Bury, and Jelen reported that enzymatic hydrolysis technology is more targeted and gentle, and it also works to change the hemicellulose and the saccharides of the cell wall [22]. However, few studies use enzymes to destroy cells in industrial

production, because an essential element of most enzymes is protein, which needs a specific environment to keep the proteins functional. Enzymes can be used to extract some of the more vulnerable substances in cells.

*3.3. Solvent Selection and Extraction Methods*

3.3.1. Solvent Selection

The efficiency and safety of the solvent are very important for the extraction of lutein. To achieve a satisfactory extraction rate, multiple extractions can be used. The polarity of the extractant should also be considered when selecting the extractant. By comparing the extraction results of several common extractants, the extraction performance of polar extractants for lutein is better than that of non-polar extractants [23].

3.3.2. Extraction Methods

Several extraction methods have been used in extracting lutein, including supercritical fluid extraction [24], solvent extraction, and flash column chromatography [25]. It is possible to change the extraction efficiency by changing the methods and the operating conditions, such as using compression, ultrasound, and microwaves [25]. Several extraction methods have been compared next.

Solvent extraction is the most common and cheapest extraction method. According to the principle of similar compatibility, when an organic solvent is mixed with algae cells, lutein will dissolve in the organic solution. Limited by the solubility of the extract, this process usually needs to be repeated many times.

Compared with traditional organic solvent extraction systems, supercritical fluid extraction (SFE) is considered to be a green technology, which has been widely used in food and drug production in recent years. There have been several reports about supercritical fluid extraction of β-carotene [26], lycopene, and other carotenoids [27]. Wu et al. (2007) reported that carbon dioxide in SFE can enter cells very quickly, so the whole extraction process is very fast [24]. There are two hydroxyl groups at the end of the lutein structure, so it has some polarity. The polarity of carbon dioxide can be adjusted by adjusting the temperature and the pressure to improve the extraction yield.

*3.4. Neurotoxicity Analysis of Solvents Used for Extraction*

Most organic solvents have varying degrees of irritation to the human body. Depending on the type, concentration, time, and frequency of exposure to organic solvents, it can cause skin allergies and also affect the central nervous system. Common organic solvents that have been found to be neurotoxic are mainly alcohols, ketones, alkanes, and benzene. The following table summarizes the neurotoxicity and related studies of some common organic solvents (Table 2).

**Table 2.** Summary of neurotoxicity of common organic solvents.

| Classification | Description | Neurological Dysfunction | Related Research |
|---|---|---|---|
| Alkanes | Hexane, the most commonly used alkane solvent, is believed to cause chronic nervous system damage. Depending on the degree, it can be partially or fully restored to healthy levels after stopping exposure [28]. Its human metabolite is 2,5-hexanedion [29,30]. | Sensorimotor or peripheral motor neuropathy, cranial and autonomic dysfunction, Sensorimotor or peripheral motor neuropathy, cranial and autonomic dysfunction [29]. | By exposing the rats to different doses of pure AZ-hexane, the rats showed the same symptoms as humans suffering from mental illness [30]. |

**Table 2.** *Cont.*

| Classification | Description | Neurological Dysfunction | Related Research |
|---|---|---|---|
| Ketones | Take *n*-butyl ketone as an example, which causes the same nerve damage pattern as hexane [28,31]. | Sensorimotor or peripheral motor neuropathy, cranial and autonomic dysfunction, Sensorimotor or peripheral motor neuropathy, cranial and autonomic dysfunction [29]. | Workers who are exposed to the compound suffer from the same type of psychosis as hexane, and it is difficult to distinguish specific types. At the same time, methyl ethyl ketone can enhance the neurotoxicity of *n*-hexane [32]. |
| Benzene | Repeated inhalation of toluene causes irreversible damage to the brain structure [33]. | Anxiety, irritability, memory loss and mood swings.Limbs and nystagmus, hearing and speech impairment, and obvious brain stem and cerebellar atrophy [33,34]. | Through multiple intravenous injections, the dog's cerebellum and cortex deformed. After the rats were exposed to toluene at a concentration of 1200–1400 pm for 14 h/day for 35 days, high-frequency hearing loss and cochlear changes were found [35]. |
| Alcohols | Ethanol mainly affects the excitability of the human body. | Ethanol can cause a decrease in nerve conduction velocity. | Low concentration of ethanol increases the excitability of the human body, increasing the concentration, the excitability decreases [36]. |

## 4. Solvent Removal

The next generation in sample preparation after extraction is usually the concentration of the extract by solvent removal. Considering the needs of large-scale production, drying is very economical and effective way. Several common drying methods are summarized as follows.

### 4.1. Freeze Drying

Dehydrated products obtained by traditional drying technology can extend the service life of food by up to one year, but traditional drying technologies may lead to a significant decrease in food quality [37]. Freeze drying is based on the principle of the sublimation dehydration of frozen products. Because there is almost no liquid water, and the temperature is low, microbial reactions virtually stop [37]. In the process of freeze-drying, solid water can protect the primary structure and shape of the product, thus improving the quality of products [37]. However, freeze drying is a very expensive method of preservation [38]. Its cost mainly depends on the material type, production cycle, and factory performance. Figure 3A,B shows a cost comparison for freeze drying between a high-value material and a low-value material [39]. It can be seen that the energy consumption of the freeze-drying process itself is negligible when dealing with high-value products [38]. Therefore, if freeze drying can add significant value to the product, or retain its high value compared with other drying methods, it may not be considered to be an expensive method of preservation.

The main operations involved in freeze drying are freezing, vacuum, sublimation, and condensation. Figure 3C shows the share of these processes in terms of total energy consumption. Note that although sublimation accounts for almost half of the total energy used, the freezing step does not consume much energy. To reduce the energy cost, any improvement to traditional freeze drying may be based on the following objectives: (1) improving heat transfer to help sublimation, (2) shortening drying time to reduce the need for vacuum, and (3) avoiding using a condenser.

### 4.2. Spray Drying

Spray drying is widely used in pharmaceutical applications [40]. It is used for the preparation of solid amorphous spray-dried dispersions (SDDs), excipient manufacture, pulmonary and biotherapeutic particle engineering, the drying of crystalline active pharmaceutical ingredients (APIs), and encapsulants [41]. Figure 4 shows the general configuration of the drying equipment used in the pharmaceutical industry [39]. To produce an SDD, a spray feed solution of API and polymer dissolved in a common solvent is usually sprayed into a hot drying gas in the chamber of the spray dryer [42]. Nitrogen is used as a drying

gas when handling organic solvents [41]. Different types of spray nozzles, including two-fluid, ultrasonic, rotary, and pressure nozzles, are usually used as required [41]. When the droplets are in contact with the drying gas, the solvent in the droplets evaporates, leaving the dry SDD particles in the drying gas. Then, they can be separated through a cyclone separator and/or bag filter [43].

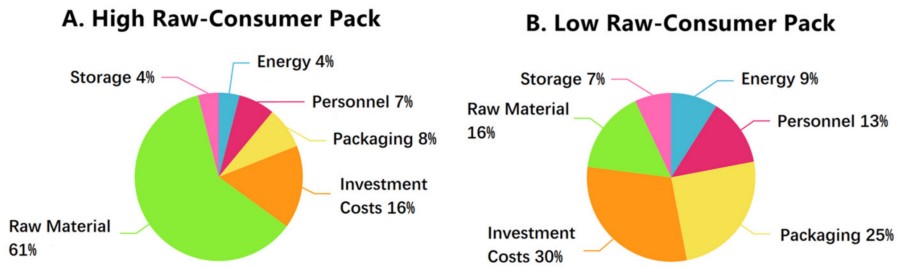

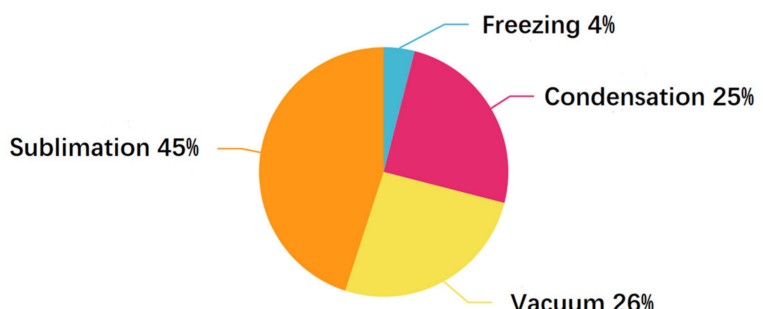

**Figure 3.** (**A**,**B**) Cost breakdown in two freeze-drying plants, processing high and low-value foods, adapted from Ref. [38]. (**C**) Energy cost breakdown for freeze-drying processes, adapted from Ref. [39].

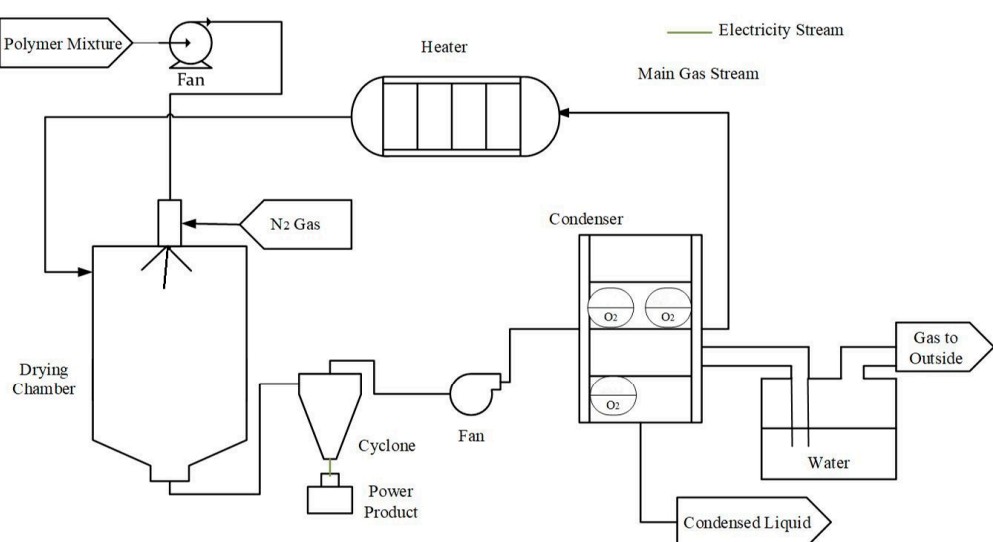

**Figure 4.** General spray-drying equipment configuration.

    The drying gas in most laboratory scale spray dryers is not recycled (open loop mode). Some spray dryers at the production scale are operated in a closed loop (or closed cycle) mode [43]. The drying gas containing the solvent is passed through a condenser, reheated, and re-introduced into the drying chamber [44]. The parameter settings for the closed loop mode are different from those of the open loop mode [43]. The flow rate of the drying gas

and the inlet and outlet temperature of the drying gas are very important parameters [45]. Dobry et al. reported that when using organic solvents, it is necessary to monitor the concentration of oxygen to prevent explosions or fires with the organic solvent [43]. The closed loop mode offers significant opportunities for process improvement, as will be discussed in Section 7 of this paper.

The thermal stability of some components is poor, so excessive temperatures may lead to the destruction of effective components. It is also very important to choose a reasonable nozzle. The smaller the droplet, the larger its specific surface area and the faster is the evaporation effect [46].

## 5. Lutein Microencapsulation and Solubility

### 5.1. Classification of Microcapsules

Microcapsules can be classified according to their size or shape, and the size of microcapsules ranges from one micron (one micrometer) upwards. However, certain microcapsules with diameters in the nanometer range are called nanocapsules to emphasize their smaller size. The morphological microcapsules can be divided into three basic types: mononuclear, multinuclear, and matrix type (Figure 5). A mononuclear core is a microcapsule with a single hollow chamber in the shell. The matrix-type are microcapsules, which have many different compounds in the shell material matrix. However, the morphology of the internal structure of the microparticles mainly depends on the shell material selected and the method for producing microcapsules [47].

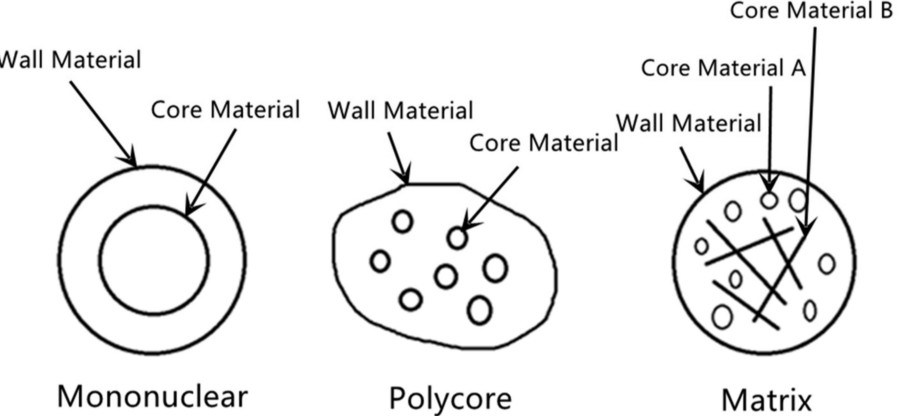

**Figure 5.** Morphology of microcapsules, adapted from Ref. [47].

### 5.2. Lutein Microcapsules

Due to the thermal instability of lutein, some other carriers are often used for mixing with lutein through spray drying to improve lutein's stability and water solubility [48]. Microcapsules can effectively protect lutein from degeneration. Commonly used single polymers (gelatin, protein, and maltodextrin) are often used as wall materials for microcapsules. Wang et al. used a mixture of porous starch and gelatin as a carrier, and lutein was wrapped to form a microcapsule structure with a core-to-wall ratio of 1:30. However, due to the low specific surface area and the weak adsorption effect of the single polymers, the lutein content in the microcapsules was low. Some preprocessing is necessary [48]. Wang et al. also added food-grade soybean phospholipids to form an emulsion [48].

Table 3 shows the main parameters of a case of lutein encapsulation used in the spray drying process. However, their use of an inlet temperature of 190 °C may be problematic when using organic solvents that have potential flammability issues. The parameters of spray drying usually vary with the size of the spray drying equipment. For scale up, a very parameter is the ratio of liquid flow rate to gas flow rate.

**Table 3.** Parameters used for the microencapsulation of lutein, adapted from Ref. [48].

| Items | Parameter |
|---|---|
| The ratio of core to wall material | 1:30 |
| Embedding Temperature | 60 °C |
| Embedding Time | 1.5 h |
| Inlet Gas Temperature | 190 °C |
| Feed Flow Rate | 50 mL/min |
| Drying Air Flow Rate | 60 m³/h |
| Encapsulation Efficiency | (94.4 ± 0.4)% |
| Yield of Product | (96.6 ± 1.7)% |
| Height | 150 cm |
| Diameter | 80 cm |

Wang et al. also measured the stability of lutein (Table 4) and measured the retention rate R (%) of lutein under different conditions (R% = $\left(\frac{C_a}{C_b}\right)$ × 100%· where $C_a$ and $C_b$ are the lutein contents before and after treatments) [48]. The information in Table 4 may be interpreted further because it contains fundamental information for scale up. A mass balance may always be written across a dryer, as follows:

$$Y_o = Y_i + \frac{L}{G}(X_i - X_o) \tag{1}$$

where Y is the humidity of the gas (kg moisture/kg dry gas), L is the flow rate of dry solids (kg dry solids/s), G is the flow rate of dry gas (kg dry gas/s), and X is the solids moisture content (kg moisture/kg dry gas). The subscripts are o for the outlet and i for the inlet. The term L $(X_i - X_o)$ is the moisture (water) that is evaporated from the solids inside the dryer. In a spray dryer, this water evaporation rate is virtually equal to the liquid fed into the dryer if the solution entering the dryer is fairly dilute, because most of the liquid fed into the dryer is evaporated, and there is typically very little moisture leaving the dryer in the outlet solids [49]. Therefore, this situation means that the liquid to gas ratio is a fundamentally useful parameter ratio for scale up from one size of a dryer to another, where both the liquid and gas flow rates should be scaled up equally to achieve the same change in humidity across the dryer. This change in humidity across the dryer is particularly important because the bulk gas humidity is important in determining the driving forces for mass transfer (drying) from the droplets and particles inside the dryer. From Table 3, the liquid to gas ratio was $5 \times 10^{-5}$, or a mass ratio of approximately 0.05 (liquid mass flow rate to gas mass flow rate). In scale up, if the mass ratio is the same, then the change in humidity across the dryer should also be the same. In addition, when scaling up, the same outlet temperature should be considered (subjects to safety considerations). The other parameter that can usefully be extracted from the data in Table 3 for scale up purposes is the air velocity through the dryer, which affects the particle residence time. In this case, the air flow rate was 60 m³/h across a dryer cross-sectional area of π/4 (0.8 m)² = 0.5 m², so the air velocity was approximately 0.033 m s⁻¹, which is a low average velocity, giving a long particle residence time of particles in the gas if the particles are micrometer-sized (1–100 µm).

The thermal stability of lutein almost certainly, like most materials, depends on the temperature of the material and the time at that temperature. The residence times in spray dryers vary from a few seconds in laboratory-scale spray dryers to minutes in full industrial-scale spray dryers. The concentration of lutein gradually decreased after both 18 h and 24 h of exposure at 120 °C [50].

**Table 4.** Comparison of the stability of free lutein and microencapsulated lutein.

| Methods (Test the Absorbance Value, λmax = 445 nm) | | Results |
|---|---|---|
| Temperature | a. 10 mL of lutein solution was heated for 10 min at different temperatures (0–100 °C). b. Keep 10 mL lutein solution at 100 °C for a certain heating time (10–60 min). | When the temperature is lower than 70 °C, heating has little effect on R (%) of free lutein and microencapsulated lutein. When the temperature exceeds 70 °C, under the same conditions, the content of free lutein decreased by 6%, while the content of microencapsulated lutein only decreased by 1% The microcapsulated lutein shows better thermal stability than unencapsulated lutein. |
| pH | Ten milliliters of lutein solution at 25 °C was tested for 1 H at different pH values (1–11). | R (%) increased in the pH range of 1–9 and decreased in the pH range of 9–11. The R (%) of microcapsulated lutein is always around 15% higher than that of free lutein during this process. |
| Light | One-hundred milliliter lutein solution at pH 7 was exposed to daylight for several days (0–30 days). | R (%) of lutein within 5 days did not change. In 5–30 days, the free lutein R (%) decreased by 43%, and the R (%) of microencapsulated lutein decreased by 7% compared with the lutein solution before spray drying. The microcapsulated lutein has better light stability than unencapsulated lutein. |
| Oxygen | One-hundred milliliter lutein solution at pH 7 was exposed for 70% oxygen content at 25 °C for a certain time (0–10 h). | Within 2 h, R (%) was relatively stable. After 2 h, the free lutein R (%) dropped to 69.4%, and the microcapsulated lutein R (%) dropped to 85.1%. The microencapsulated lutein has better oxygen stability than unencapsulated lutein. |

R (%) means the retention rate.

### 5.3. External Morphology and Glass Transition Temperature of Maltodextrin-Lutein Microcapsules

Kuang et al. (2015) prepared three mixed solutions of maltodextrin with sucrose at three different mass ratios (3:0, 3:1, 3:3) and added soybean phospholipids to obtain an emulsion, which was combined with lutein. Then, the emulsion was spray dried to obtain microcapsules of lutein [51].

The external morphology of the microcapsule lutein was further studied, and it was found that the external morphology was relatively complete, with no visible cracks or holes, but there was a certain degree of collapse. The higher the mass fraction of sucrose in the emulsion, the lower the degree of surface collapse and the higher the sphericity. Through the research of Kim, Chen, and Pearce, the migration of water from the inside to the surface occurs most significantly in the first period of spray drying, the unhindered drying period (sometimes called the constant rate drying period, although this rate is only constant if the external conditions are constant) [52]. A large amount of water evaporates at this time, and the internal water continues to migrate outward. Due to the presence of multiple double bonds in the lutein structure, lutein has strong hydrophobicity, which makes the internal lutein easier to migrate to the surface. Comparing D and F in Figure 6, washing off the surface lutein has no substantial effect on the morphology of the particles. The purpose of washing away the free lutein on the surface is to more accurately measure the lutein content inside the particles [51].

The glass transition temperature of lutein microcapsules is important, because Bhandari et al. have shown that this particle property affects the stickiness and wall deposition tendency of spray-dried materials, which in turn affects their solids recovery in spray drying and the flowability of the particles. This glass transition temperature decreases with the increase in the sucrose mass fraction. The molecular weight distribution of the wall material can be changed by changing the proportion of sucrose in the wall material [51]. Table 5 lists the initial glass transition temperature, the transition width of the wall materials with different mass ratios, and the trend in the color parameters caused by the change in the sucrose mass fraction.

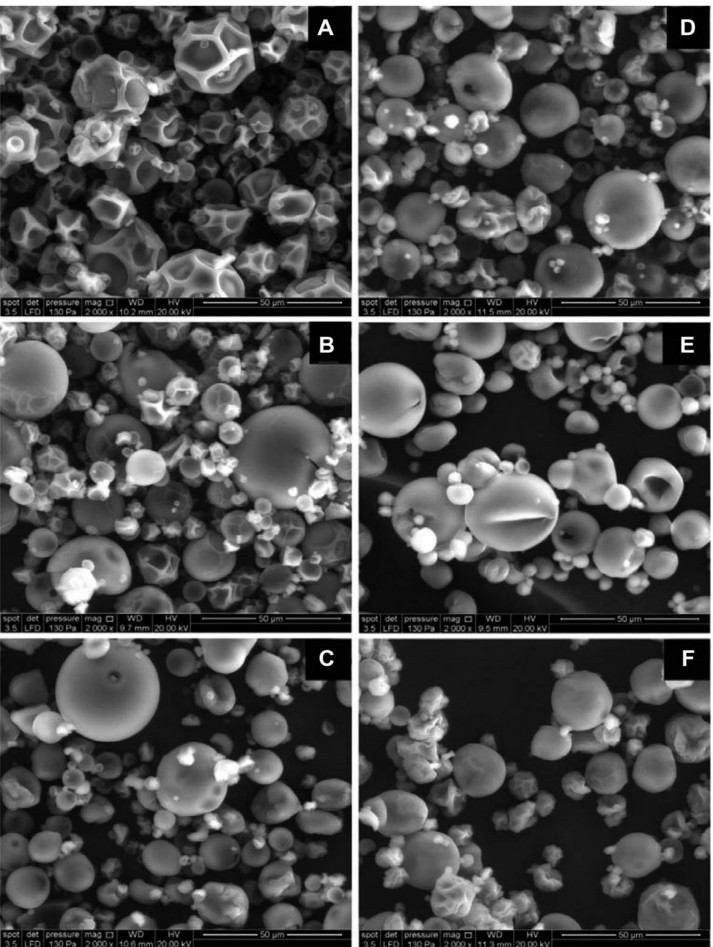

**Figure 6.** SEM of selected lutein microcapsules: (**A**) M040:0 (pure maltodextrin M040), (**B**) M100:0 (pure maltodextrin M100), (**C**) M180:0 (pure maltodextrin M180), (**D**) M040:1 (weight ratio of maltodextrin M040: sucrose = 3:1), (**E**) M040:3 (weight ratio of maltodextrin M040: sucrose = 3:3), and (**F**) M040:1 (weight ratio of maltodextrin M040: sucrose = 3:1, washed with hexane to remove the surface lutein), adapted from Ref. [51].

**Table 5.** The glass transition temperature of lutein-containing particles and the trend in this temperature with changing sucrose content, adapted from Ref. [51].

| Microcapsules | Tgi (°C) | (Tge−Tgi) (°C) | L* | a* | b* |
|---|---|---|---|---|---|
| M040:0 | NA | NA | 64.9 ± 0.89a | −2.16 ± 0.11d | 29.8 ± 0.51f |
| M040:1 | 82.4 ± 0.36b | 23.0 ± 1.69bc | 60.6 ± 0.36def | −1.42 ± 0.01a | 36.6 ± 0.17cd |
| M040:3 | 65.1 ± 0.16d | 20.3 ± 2.66cd | 62.0 ± 0.95cd | −1.36 ± 0.10a | 40.0 ± 0.56b |
| M100:0 | NA | NA | 64.2 ± 0.59ab | −2.28 ± 0.06d | 32.1 ± 0.25e |
| M100:1 | 76.1 ± 2.00c | 25.9 ± 1.45b | 60.0 ± 0.48ef | −2.80 ± 0.08e | 31.8 ± 0.23e |
| M100:3 | 59.2 ± 0.28e | 18.6 ± 0.29cd | 64.1 ± 0.89ab | −1.99 ± 0.04c | 37.4 ± 0.50c |
| M180:0 | 106.1 ± 1.70a | 18.4 ± 3.39cd | 62.6 ± 1.71bc | −2.21 ± 0.09d | 36.2 ± 0.86d |
| M180:1 | 73.5 ± 2.96c | 32.4 ± 3.59a | 59.3 ± 1.70f | −1.40 ± 0.06a | 41.1 ± 1.08a |
| M180:3 | 56.7 ± 0.28e | 17.3 ± 0.39d | 61.3 ± 0.90cde | −1.78 ± 0.01b | 39.0 ± 0.56b |

Values represented the mean ± standard deviation, and values that were followed by different letters within each column were significantly different ($p < 0.05$). Tgi, onset glass transition temperature; (Tge-Tgi), glass transition temperature width; NA, not available. L, a, b are the color parameters obtained by the Minolta colorimeter device; L* indicates lightness, a* is the red/green coordinate, and b* is the yellow/blue coordinate; L* (L* sample minus L* standard) = difference in lightness and darkness (+ = lighter, − = darker); a* (a* sample minus a* standard) = difference in red and green (+ = redder, − = greener); b* (b* sample minus b* standard) = difference in yellow and blue (+ = yellower, − = bluer), adapted from Ref. [53].

The microencapsulation of lutein improves the stability of lutein through many aspects. The core material is wrapped in the center of the particles, and wall materials in different proportions are used to obtain different physical and chemical properties. Wang et al. selected maltodextrin as the microencapsulating wall material to improve the stability of lutein in the presence of higher temperatures, different pHs, and oxygen [48]. At the same time, by controlling the content of sucrose, the surface shape of the microcapsule particles was affected.

Different encapsulating materials have significantly different properties. Starch can be used to enhance the stability of flavoring agents due to its emulsifying properties during drying [54]. Chitosan is a polysaccharide that is soluble in acidic aqueous solutions. It has very good biocompatibility and is often used for the encapsulation of drugs [55]. Ascorbic acid can reduce moisture absorption during the spray drying process, and it does not easily agglomerate during the whole process [56]. Chiou and Langrish produced *H. sabdariffa* L.- Citrus Fiber microcapsules by using spray drying. A citrus powder containing biologically active substances and suitable for water storage was obtained. It was found that citrus powder is an alternative to maltodextrin as another wall material [57].

### 5.4. Characterization of Encapsulated Lutein

Encapsulation content, encapsulation efficiency, particle size, water activity, and moisture content are important basic parameters of encapsulated products. These parameters determine the spoilage time of the product. Through the different wall materials, these basic characteristics will change. For example, consider Ding et al.'s study as a study of lutein stability by using different carbohydrates as wall materials (Table 6) [58].

**Table 6.** Characterization of carbohydrate microencapsulation of lutein, adapted from Ref. [58].

| Encapsulation Material and Mass Ratio | Median Particle Size (um) | Moisture Content (%) | Moisture Adsorption (%) | Encapsulation Efficiency (%) | Retention Value (%) | Product Yield (%) |
|---|---|---|---|---|---|---|
| Sucrose | 7.0 ± 0.5 | 1.4 ± 0.2 | 0.8 ± 0.3 | 0.7 ± 0.5 | 22.2 ± 0.7 | 55.6 ± 1.4 |
| Trehalose | 7.6 ± 0.5 | 3.3 ± 0.3 | 9.0 ± 0.3 | 70.6 ± 1.2 | 86.3 ± 2.2 | 69.1 ± 3.2 |
| Inulin | 7.7 ± 0.6 | 3.8 ± 0.5 | 8.4 ± 0.3 | 75.0 ± 0.7 | 86.5 ± 0.9 | 67.7 ± 3.1 |
| Modified starch | 9.4 ± 0.7 | 2.3 ± 0.3 | 9.4 ± 0.4 | 73.2 ± 0.9 | 83.6 ± 0.9 | 92.6 ± 1.0 |
| Maltodextrin 10 | 7.5 ± 0.6 | 3.5 ± 0.2 | 12.4 ± 1.3 | 61.2 ± 1.1 | 84.1 ± 4.5 | 92.4 ± 1.2 |
| Maltodextrin 15 | 7.2 ± 0.4 | 3.5 ± 0.2 | 13.6 ± 1.0 | 58.9 ± 1.8 | 82.4 ± 2.9 | 90.7 ± 1.8 |
| Maltodextrin 20 | 6.8 ± 0.8 | 3.1 ± 0.2 | 16.7 ± 0.6 | 56.1 ± 2.8 | 83.3 ± 2.8 | 83.1 ± 1.7 |

Considering this case, the moisture content after encapsulation is determined by the feed rate, inlet/outlet temperature, and other processes. The residual water content was less than 4% (dry basis), which is an ideal edible powder. The moisture content of modified starch is lower than that of sucrose, possibly because sucrose has more exposed hydroxyl groups than modified starch, which made the sucrose more hygroscopic and holding more water during the spraying process [59]. The hygroscopicity of the seven encapsulated powders varied from 1 to 17%. The higher hygroscopicity of lutein-maltodextrin microencapsulation powder was due to the higher hygroscopicity of maltodextrin itself. Fernandes et al. (2014) also pointed out that using inulin to partially replace maltodextrin can reduce the water absorption of the microencapsulated powder [60].

Comparing the encapsulation efficiency and retention value, sucrose is significantly worse than trehalose, which may be due to the simpler formation of single crystals during the drying process, which cannot combine with other chemical components. Suryabhan et al., reached the conclusion that: The lower the glucose equivalent in maltodextrin, the higher the encapsulation efficiency, because the glass transition temperature of maltodextrin is decreased by higher glucose equivalents [61].

### 5.5. Characterization of Encapsulated Lutein

Muhoza et al. encapsulated lutein into glycosylated casein and conducted controlled release experiments in a simulated gastric solution [62]. The simulated gastric fluid con-

tained 7 mL hydrochloric acid, 2 g sodium chloride, 3.2 g pepsin (activity 3000–3500 U/mg), and 250 mL deionized water. Then, 1 mol/L hydrochloric acid and deionized water were used to adjust the pH to 1.2 and the volume to 1 L. The simulated intestinal solution contained 0.68% (*w/v*) $KH_2PO_4$, 0.062% (*w/v*) NaOH and 10 mg/mL pancreatin (285 U/mg protease, 56 U/mg lipase, 288 U/mg amylase). Figure 7A shows the cumulative release of lutein micelles in simulated gastric solution for three hours. In the first half an hour, lutein was rapidly released from the simulated gastric solution and reached more than 50% of the final amount. Figure 7B shows the relationship between the average diameter of lutein micelles and the reaction time. The average particle size of lutein micelles increased from 470 nm to 1230 nm.

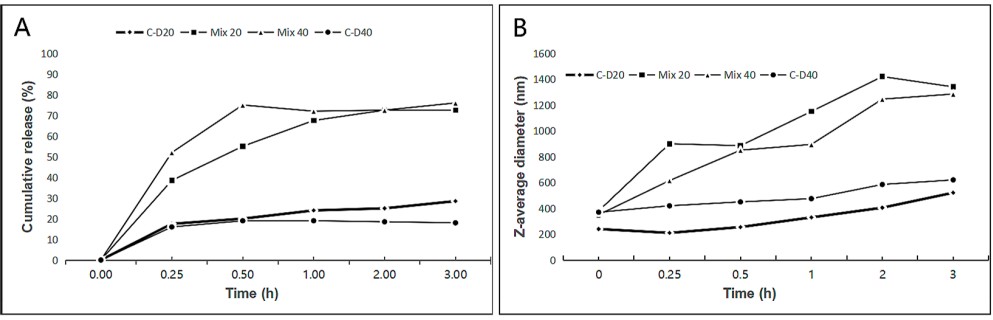

**Figure 7.** Controlled release of lutein micelles in simulated gastric juice (**A**) cumulative release of lutein micelles in simulated gastric solution for three hours; (**B**) relationship between the average diameter of lutein micelles and the reaction time, adapted from Ref. [62].

## 6. Gap Analysis and Opportunities for Process Improvement

There are two main directions for improvement in this field in the future: microencapsulation technology and sustained release of lutein. Energy quality analysis (exergy, availability) is also a consideration.

Microencapsulation technology improves the various stability of lutein and has very broad market prospects. However, the encapsulation materials of the current microencapsulation technology are mainly single substances, such as maltodextrin and emulsified starch [63], or a mixture of two pure substances to increase the encapsulation efficiency. The biologically active citrus powder produced by Chiou and Langrish may also be suitable for encapsulating lutein. Plant powders have natural porous structures and may absorb the lutein [57].

Another direction is to produce oil-in-water emulsions with a controlled release function to improve the bioavailability of lutein. Incorporating lutein into the surfactant and lipid system is a useful possibility to consider because lipids can delay the emptying of the gastrointestinal tract [64], and lutein will have a longer time for absorption. Muhoza et al. encapsulated lutein in glycosylated casein micelles to release them slowly [62]. Lutein can also be incorporated into a polymer matrix to synthesize polymer microspheres with controlled release properties, thereby increasing the absorption efficiency of lutein in the human body as potential functional food [65].

As suggested in Section 4.2 on spray drying, the closed loop system for spray drying has advantages and disadvantages compared with the more common open loop configuration. Advantages include handling flammable solvents with much greater safety, better containment for sensitive or toxic materials, and potential energy savings (and operating cost savings) through recycle of energy-containing gases. Disadvantages include greater process complexity meaning greater capital cost, and some greater needs for cleaning the recycled gases. Managing the challenges posed by these disadvantages and maximizing the benefits from the advantages is a key consideration in the process engineering involved in the production of microparticles, including microencapsulated lutein.

## 7. Conclusions

In this review, the nutritional value of lutein and its adjuvant treatment value for diseases have been summarized, and predictions have been made for the market prospects of lutein as a functional food. The review has discussed the selection of algae species, the pretreatment of algae, the destruction of algal cell walls, and the selection of organic solvents, as well as the use of the closed-loop spray drying technology as a very suitable technology to microencapsulate lutein. Microencapsulation technology may become a key development direction for the production of potentially unstable dietary supplements in the future. The advantages and disadvantages of different microcapsule wall materials and the influence of glass transition temperature on the microcapsulation process have been analyzed and compared. At the same time, some examples of improving the stability of lutein in various ways through microencapsulation technology have been discussed. Opportunities for further research that can be pursued in the future include increasing the content of lutein bound in microcapsules and testing the digestion and absorption mechanisms of microencapsulated lutein in different in vitro and in vivo systems and models of the human gastrointestinal tract.

**Author Contributions:** Z.L.: Conceptualization, Writing—original draft. T.L.: Conceptualization, Writing—review and editing, supervision. All authors have read and agreed to the published version of the manuscript.

**Funding:** This research received no external funding.

**Institutional Review Board Statement:** Not applicable.

**Informed Consent Statement:** Not applicable.

**Conflicts of Interest:** The authors declare no conflict of interest.

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
