# Peer review of "A Review of the Extraction and Closed-Loop Spray Drying-Assisted Micro-Encapsulation of Algal Lutein for Functional Food Delivery"

_processes, doi:10.3390/pr9071143_

Round 1
Reviewer 1 Report
Manuscript “A Review of the Extraction of Lutein from Algae and its Micro-encapsulation Based on a System as a Functional Food Delivery System” by Zexin Lei and Timothy Langrish presents an overview of algal lutein extraction and microencapsulation. Lutein is considered as an important nutraceutical with wide use in human and domesticated animal nutrition. Significant efforts were made in fundamental and applied sciences to enhance lutein production and improve final product formulation. Presented review covers lutein extraction and microencapsulation reasonably well. All other topics e.g. market value and biological effects, emphasized in article’s abstract and conclusions are not developed well or even contain misleading information. Therefore, they have to be omitted from listing as reviewed and aggregated in introduction section after appropriate corrections.
Additional comments
Title
- Title is somewhat confusing, especially, its last part. Modifications to title can be made e.g. “extraction and Closed-loop Spray Drying-assisted Micro-encapsulation of Algal lutein for a functional food delivery”
Abstract
- There is a small paragraph in the manuscript’s body devoted market value of lutein. It is hard to characterize it as “The potential market value of lutein has been evaluated.”
Main text
- Lutein Introduction: Following sentence should be extended with examples and references or deleted “It also has some benefits in other aspects of human health.”
- “immune labeling” correct to “immunolabeling”
- Many sentences have to be rephrased e.g. “Lutein and algae are bioproducts that must be processed to produce bio-available products.” Despite the vague use of the term bioproducts it is difficult to say that algae are bioproducts.
- The introduction is not general. Just several research papers are cited with most of the comprehensive reviews omitted. Also, it is very difficult to find logic in a presentation. There is jumping from topic to topic.
- Benefits of Lutein: Formatting this section’s text into table 1 is not helpful. There is no need for such table taking in account singular examples of possible lutein involvements in physiological and pathological events. Reformatting of this table content into regular text can address such an issue.
- There is no information on the lutein market in this section. Is Beta Carotene Market mentioned? Lutein is not part of this market. What is the size of the carotenoid market and what portion of lutein in this market?
- Is table 2 just a reproduction of a table published in ref [15] or some modifications are made? This information should be added and, if needed, permissions received. The same is true for most other Figures and tables. Reproduction of figures and tables without permissions is not acceptable I most of cases. For example, Figure 3A is recolored version of figure presented in ref [29] with added unnecessary legends and duplicated % values, again, unnecessary.
- “4.2. Selection of Algae” is a wrong title for corresponding section.
- It is not clear why Data presented in fig 2 can not be imputed in table 3. This will make information presentation more concentrated and useful.
- Figure 6 has no marks on Y and X axes making presented dimensionality useless. E.g. Time in seconds can be milliseconds and millions of seconds.
- More information should be added into Fig. 7 legend including a description of used abbreviations.
Author Response
Thank you very much for your suggestions, they are very valuable and I have accepted them all. For details, see the response file and the new manuscript of the paper.

Reviewer 2 Report
General question is; are Tables and Figures adapted strait from the references cited in the each case (as e.g. Figure 1 from ref 17– this Figure is the same as in the Ref 17, where it is entitled “Systemic ultrastructure of C. vulgaris representing different organelles” ) – have Authors permissions to present not changed figures and Tables – if yes (I hope it is) – it should be in each case noted “with permission”.
In Introduction and in whole manuscript English should be more concise and should be corrected.
Line 44; It should begin as follows; “Recently, Becerra et al. have…”
Line 55; “previously presented in the previous literature” – “previously” should be removed.
In the Table 1 “Items” should be replaced by “Activity of Lutein”, and in the column; “Cancer” replaced by “Anti-cancer”, Eye diseases” by “Eye diseases prevention”, next page in this column; “cardiovascular diseases” should be “Application in the cardiovascular diseases”.
Lines 64-69; What is the reason to section “3. Lutein market” – in this section Authors mentioned about Lutein nutraceuticals or about nutraceuticals at all – not clear.
Line 97 ; this line should ends as follows; “…of chlorella cell [17].”
Systematic names of the species (in the main text e.g. lines 117 and 126; in the captions for figures ) should be written in italics (e.g. Chlorella vulgaris Fig 1 and Fig 2).
Lines 55, and 146-151 – English should be more fluent and concise – repetitions should be avoided.
Line 154-155; “Solution extraction” – should be “Solvent extraction” (?)
Lines 168-173 – not clear
Line 174 – “Solvent removal” – it is not Extraction method. It should be entitled e.g. “Sample concentration methods” or section “5. Solvent removal”; It will be helpful; “The next of the sample preparation is concentration of the extract by solvent removal”
After section 4.4.1. Authors placed section “5.1 Freeze drying” (line 178) and next in the Line 204 is again “5.2 Freeze drying” – this section is repeated twice and no application for Lutein are presented in this section/sections
Table 5 is not consistent; Lines about Temperature should be as follows; “10 mL of lutein solution was heated for…” and “10 mL of lutein solution was kept at 100…” in the pH; “10 mL of lutein solution at 25 oC was tested for 1 H at different pH values (1-11).”
Line 298; citation needed.
Figure 7; (D)M040:1 and (F)M040:1 – repetitions shows different pictures (?)
Figure 8 is not very informative.
Author Response

(The authors gave the same response as above.)

Reviewer 3 Report
Dear Authors,
It is a good review. I have some comments.
- I suggest to introduce the chemical structure of lutein, and comment briefly about the molecule.
-Lutein market. The above is interesting, however the information provided is about nutraceuticals and carotene market. Is there any information focused in lutein market? If yes, comment about it.
- References are not in accordance with journal style. Check it.
Author Response

(The authors gave the same response as above.)

Reviewer 4 Report
The present manuscript summarize the state-of-art of lutein extraction procedures, mainly from algae, and its microencapsulation for food supplement purposes.
The review of literature data is pretty well organized and conclusions supported by the existing knowledge. On the whole, the manuscript deserves its publication in the journal Processes, but few issues should be addressed:
- Introduction should be better organized in a more logic manner, in support of the need of lutein extractions from algae.
- Tables should be cited also in the text (i.e. table 1).
- The first phrase of chapter 4.1 should be deepened, explaining the role of growth rate, lutein/zeaxanthin ratio, chlorophyll A/lutein ratio, etc.
- A chapter discussing toxicological issues related to the extraction procedures would rise the appeal of the manuscript, since authors are dealing also with food supplements.
Author Response

(The authors gave the same response as above.)
